# Oxygen Vacancy-Mediated Activates Oxygen to Produce Reactive Oxygen Species (ROS) on Ce-Modified Activated Clay for Degradation of Organic Compounds without Hydrogen Peroxide in Strong Acid

**DOI:** 10.3390/nano12244410

**Published:** 2022-12-10

**Authors:** Tianming Wu, Jing Cui, Changjiang Wang, Gong Zhang, Limin Li, Yue Qu, Yusheng Niu

**Affiliations:** 1Institute of Biomedical Engineering, College of Life Sciences, Qingdao University, Qingdao 266071, China; 2College of Resources and Environment, Shandong Agricultural University, Taian 271018, China; 3Shandong Zhengyuan Geological Resource Exploration Co. Ltd., China Metallurgical Geology Bureau, Weifang 261200, China; 4Center for Water and Ecology, State Key Joint Laboratory of Environment Simulation and Pollution Control, School of Environment, Tsinghua University, Beijing 100084, China; 5School of Tourism and Geography Science, Qingdao University, Qingdao 266071, China

**Keywords:** advanced oxidation process, strong acid wastewater, reactive oxygen, organic compounds, electron transfer

## Abstract

The treatment of acid wastewater to remove organic matter in acid wastewater and recycle valuable resources has great significance. However, the classical advanced oxidation process (AOPs), such as the Fenton reaction, encountered a bottleneck under the conditions of strong acid. Herein, making use of the oxidation properties of CeAY (CeO_2_@acid clay), we built an AOPs reaction system without H_2_O_2_ under a strong acid condition that can realize the transformation of organic matter in industrial wastewater. The X-ray photoelectron spectroscopy (XPS) proved that the CeAY based on Ce^3+^ as an active center has abundant oxygen vacancies, which can catalyze O_2_ to produce reactive oxygen species (ROS). Based on the electron spin-resonance spectroscopy spectrum and radical trapping experiments, the production of •O_2_^–^ and •OH can be determined, which are the essential factors of the degradation of organic compounds. In the system of pH = 1.0, when 1 mg CeAY is added to 10 mL of wastewater, the degradation efficiency of an aniline solution with a 5 mg/L effluent concentration is 100%, and that of a benzoic acid solution with a 100 mg/L effluent concentration is 50% after 10 min of reaction. This work may provide novel insights into the removal of organic pollutants in a strong acid water matrix.

## 1. Introduction

Organic wastewater produced in industrial and living areas has complex components, a high chroma, and a pungent smell, which is easy to cause water quality deterioration [1,2]. As a well-known representative of organic wastewater, dye wastewater contains abundant refractory compounds [3,4]. The pH of mother liquor production in many acidic dye wastewaters is less than 2.0, which needs lots of bases to adjust the pH and makes it tough to treat by traditional approaches such as chemical and activated sludge methods. This will cause a waste of resources and is not conducive to sustainable development [5,6,7]. The recovery of acid from strong acid industrial wastewater has great economic benefits and promotes the reuse of resources [8]. Using appropriate technology to eliminate the pollution of acid wastewater and reuse valuable resources is the key to acid wastewater treatment [9,10]. The yearly production of dyes in China is 90,000 tons, or 60% of the world’s. During the production, 2% of the dyes flowed into the water and polluted the environment seriously [11,12]. Therefore, it is urgent to develop reasonable methods to handle organic wastewater under strongly acidic conditions.

In recent years, many wastewater treatment processes have been developed, especially the advanced oxidation process (AOPs) [13]. Fenton oxidation technology, a classical AOPs, which is widely used in dye wastewater treatment [14]. In the Fenton reaction, Fe^2+^ activates hydrogen peroxide (H_2_O_2_) and leads to the generation of various reactive oxygen species, including singlet oxygen (^1^O_2_), hydroxyl radicals (•OH), and superoxide anion radicals (•O_2_^−^), which can degrade organics under acidic conditions [15,16,17]. However, the optimal Fenton reaction has a narrow range of applications [18,19,20]. The efficiency of the Fenton reaction will be reduced in strongly acidic conditions. It is necessary to adjust the pH value before and after treatment under strong acid conditions. Therefore, the traditional Fenton reaction encountered a bottleneck that its narrow application range of pH is about 3.5–4.0 and H_2_O_2_ disproportionation might occur under strong acid conditions. In addition, the disproportionation of H_2_O_2_ and the production of iron precipitation also limit the normal operation of the Fenton method [18,21]. Therefore, the effective degradation of organic compounds in the environment without H_2_O_2_ addition under strong acid conditions is a key problem.

The excellent redox properties (cycling between Ce^3+^ and Ce^4+^), presence of oxygen vacancies, and oxygen transport/storage ability of cerium oxide (CeO_2_) make it attract significant attention in diverse fields such as biology, oxygen sensors, solar cells, and catalysis applications [22,23,24]. The Ce^3+^ in CeO_2_ creates an electron imbalance, forming oxygen vacancies and unsaturated chemical bonds, which contribute to the formation of chemisorbed oxygen on the catalyst surface [25,26,27]. The chemisorbed oxygen and oxygen vacancies will create the superoxide species for the degradation of the organic compounds [28,29]. However, the performance of a single CeO_2_ is still unsatisfied. An important factor restricting the application of CeO_2_ is its low specific surface area and a low ratio of Ce^3+^/(Ce^3+^ + Ce^4+^) [30].

In this work, to resolve the degradation of organic matter in strong acid wastewater, avoid agglomeration of CeO_2_ microspheres, and increase the proportion of Ce^3+^ in (Ce^3+^/Ce^4+^), a CeO_2_-based catalyst for degradation of organic pollutants under strong acid conditions was designed. When CeO_2_ was loaded on activated clay, CeO_2_ microspheres are uniformly distributed on the surface of activated clay, the specific surface area of the material was increased, and the proportion of Ce^3+^ in (Ce^3+^/Ce^4+^) also increased. In the system of pH = 1.0, when 10 mg CeAY is added to 10 mL of wastewater, the degradation efficiency of an aniline solution with a 5 mg/L effluent concentration is 100% and that of a benzoic acid solution with a 100 mg/L effluent concentration is 50% after 10 min of reaction. The X-ray photoelectron spectroscopy (XPS) proved that the CeAY based on Ce^3+^ as an active center has abundant oxygen vacancies, which can catalyze H_2_O to produce reactive oxygen species (ROS). Based on the electron spin resonance (ESR) spectroscopy spectrum and radical trapping experiments, we determined the production of •O_2_^−^ and •OH, which are important factors of degradation in the organic compounds, which provides an effective support for CeAY to degrade organic compounds under strong acid conditions. The CeAY can degrade organics in strong acidic wastewater without adding hydrogen peroxide, breaking the limit of traditional Fenton reaction pH values, saving resources for pH adjustment, and facilitating in situ acid recovery, which is of great significance for saving resources and protecting the environment.

## 2. Materials and Method

### 2.1. Materials

Ce(NO_3_)_3_•6H_2_O (99.5%, metals basis), Na_3_C_6_H_5_O_7_•2H_2_O (99.5%, metals basis), hydrochloric acid (HCL), Sulphuric-acid (H_2_SO_4_), sodium hydroxide (NaOH), Formic acid (FA), Rhodamine B (Rh B), Methyl Orange (MO), Aniline (A) and Benzoic acid (BEN) was supplied by Shanghai Macklin Biochemical Co., Ltd. (Shanghai, China). Bentonite clay was purchased from Hebei Xincheng Mining Co., Ltd. (Shijiazhuang, China). All reagents were used without further purification and Milli-Q ultrapure water (18.2 MΩ·cm) was used for the preparation of aqueous solutions.

### 2.2. Synthesis of Activated Clay CeO_2_ and CeAY

#### 2.2.1. Synthesis of Activated Clay

The synthesis of activated clay is through inorganic acid heating. The purchased bentonite is added to 5% H_2_SO_4_, then heated and stirred at 90 °C for 2 h, then the material is washed to neutral with deionized water, then dried at 60 °C, ground into powder, and placed in a 50 mL centrifuge tube for standby. The composition of Active clay is shown in Appendix A.

#### 2.2.2. Synthesis of CeO_2_ and CeAY

A hydrothermal procedure was employed to synthesize CeAY. Add 0.6 g of Ce (NO_3_)•6H_2_O into 10 mL of deionized water, stir it magnetically for 10 min to dissolve it, and then add it dropwise into a beaker containing 3.2 g of Na_3_C_6_H_5_O_7_•2H_2_O and 3.2 g of sodium hydroxide under continuous stirring. After stirring for 30 min, add 0.6 g of activated clay and stir for 30 min. After stirring for 1 h, the precipitate was transferred to a Teflon-lined stainless steel autoclave and maintained at 120 °C for 24 h. After cooling to room temperature, filter the precipitate, wash it with water and ethanol alternately several times, and then dry it in an oven at 60 °C overnight. Grind it into powder with a mortar and collect it in a centrifuge tube.

### 2.3. Degradation Tests

The catalytic activities of the prepared materials were evaluated based on the degradation of Rh B, MO, aniline, and benzoic acid in aqueous solutions at room temperature. The Rh B and MO degradation were performed in 10 mL Beaker containing 5 mL of the Rh B and MO solutions. The initial solution pH was 1.0 and Rh B, MO concentration of 2 mg/L, an aniline concentration of 10 mg/L, and benzoic acid concentration of 100 mg/L. In a typical test, a fixed amount of catalyst was added to the solution and has a vortex to achieve fully mixed. At fixed intervals, 3 mL samples of the solution were removed with a syringe and quenched with excess methanol to prevent further reaction. The suspensions were separated by centrifugation at 3000 r/min for 5 min to obtain the supernatant, and filter through 0.22 µm filter membrane for subsequent analysis.

### 2.4. Investigation of the Oxidase Mimetic Activity of CeAY

The oxidase-mimic behavior of the CeAY was measured based on the oxidation of TMB without H_2_O_2_. In short, 50 μL of CeAY (0.04 mg mL^−1^) and 10 μL of TMB (0.005–8 mM) were incubated in HCl 0.1mol L^−1^ for 10 min for absorbance collection.

The catalytic activity of the CeAY was evaluated within different concentrations of TMB for kinetics study. The apparent kinetic parameters were measured based on the following Michaelis equation:(1)$$\;V=\frac{V_{max}\times \left[S\right]}{K_{m}+\left[S\right]}$$
where [*S*], *K_m_*, *V*, and *V*_max_ are the concentration of the substrate, the Michaelis constant, the initial velocity and the maximal reaction velocity, respectively. *K_m_* and *V*_max_ were calculated based on the Lineweaver–Burk plot strategy.

### 2.5. Characterization

The UV absorption was carried out with a Thermo Scientific Multiskan FC (Shanghai, China). The microstructure and elemental mapping analysis were characterized by scanning electron microscopy (SEM, JEOL JSM-6500FE, Tokyo, Japan). X-ray photoelectron spectroscopy (XPS) data were obtained by a PHI5000 Versaprobe-II spectrometer with a monochromatic Al Kα (1486.6 eV) source (Japan). The surface functional groups were analyzed by Fourier transform infrared (FTIR) spectroscopy using a Thermo Scientific Nicolet iS10 FT-IR spectrometer (Thermo Fisher Scientific INC, Waltham, MA, USA). X-ray diffraction (XRD) characterizations were performed using a Bruker D8 Advance diffractometer with a copper Kα (λ = 0.154056 nm) radiation source (Germany). The room temperature electron spin resonance (ESR) spectra were obtained by using a Bruker A300 EPR electron paramagnetic resonances spectrometer (Karlsruhe, Germany). The degradation of aniline and benzoic acid were performed using an Agilent 1260 LC (Santa Clara, CA, USA). The chromatographic separation of benzoic acid was achieved with a C_18_ column and isocratic elution, with a mixture of acetate buffer (pH = 4.4) and methanol (65:35) as the mobile phase, the effluent was monitored at 235 nm. The chromatographic separation of aniline was achieved with a C_18_ column and isocratic elution, with a mixture of methanol and water (75:25) as the mobile phase, the effluent was monitored at 270 nm.

## 3. Results and Discussion

### 3.1. Characterization

The microstructures and morphologies of the samples were first characterized by scanning electron microscope (SEM), as the representative SEM images of the CeAY samples illustrated in Figure 1 and Appendix A. From these graphs, one can see the spherical structure of CeO_2_ was agglomerated, and the diameter was about 20–30 nm (Appendix A). The surface of acid clay exhibited the layered structures, and the size is about 3 µm (Appendix A). In the process synthesis of CeAY, in order to avoid CeO_2_ agglomeration and affect the catalytic, CeO_2_ was loaded on the activated clay by in-situ precipitation, as depicted in Figure 1b, the CeO_2_ was averagely anchored on the surface of Acid Clay, the size of CeAY become 3–5 µm, and the surface was filled with rice shaped CeO_2_ with a diameter of approximately 50 nm (Figure 1b). From the energy dispersive spectroscopy (EDS) analysis, the CeAY was identified as having cerium, silicon, and aluminum, and these elements are evenly distributed in the material (Figure 1e and Appendix A). The results indicated that the CeO_2_ was successfully and evenly distributed and loaded onto the acid clay surface.

The N_2_ gas adsorption method at 77 K was used to explore the surface area of the CeAY, and the multi-point Brunauer-Emmett-Teller (BET) method was used to investigate the surface area analysis was performed, and the result is 94.463 m^2^/g. This value is about 37.61% more than that of acid-treated clay and about 36.133 times that of CeO_2_ (Appendix A). Appendix A shows the adsorption and desorption of N_2_ (i.e., the adsorption isotherm) on the surfaces of CeO_2_, acid-treated clay, and CeAY at 77 K. The adsorption isotherm of type I was found on active clay, indicating multilayer adsorption. The CeO_2_ and CeAY follows Type IV isotherm with a hysteresis behavior, this can be attributed to the adsorption/desorption behavior in the capillary condensation adsorbent and indicating that the size of CeO_2_ and the surface of CeAY was about to nm.

The XRD patterns of as-prepared CeO_2_ powder, acid-treated clay, and CeAY by the hydrothermal method are shown in Figure 1c. The phase and purity of the as-synthesized samples were used XRD analysis to investigate. The reflection peaks at (111), (200), (220), (311), (222), (400), (311), (420), and (422) are assigned to the crystal planes’ face-centered cubic structure of CeO_2_ (JCPDS Card No. 34-0394) [31]. Additionally, the acid-treated clay’s reflection peaks can be assigned to the SiO_2_ reported in JCPDS Card (NO.39-1425), where the peak of 2θ = 19.066° is the principal peak of SiO_2_, and the peak of 26.586° is the principal peak of Al_2_SiO_5_, indicating that the main components of acid-treated clay are SiO_2_ and Al_2_SiO_5_ (Appendix A) [32]. After CeO_2_ was precipitated on the acid-treated clay, the reflection peaks included the two principal peaks of CeO_2_ and acid-treated clay, and the major peak of 2θ = 31.384° was corresponding to the Al_2_Ce(220) (JCPDS No. 25-1120), while the principal peak of 2θ = 27.619° was responding to the (021) of Ce_2_SiO_5_ (JCPDS No. 40-0036), reconfirming that the CeO_2_ was successfully added to the acid-treated clay (Appendix A). From Figure 1c, we can see that the reflection peak of the CeAY has characteristic peaks of both CeO_2_ and acid-treated clay, further indicating that the CeO_2_ was successfully added to the acid-treated clay.

Fourier transform infrared spectroscopy (FTIR) is an excellent characterization for embellished groups. Figure 1d presents the FTIR spectra of the CeAY catalyzer before (Curve a) and after adding sulfuric acid (Curve b) and MO (methyl orange) degradation (Curve c). The absorbance peak at 3400 cm^−1^ is attributed to the O-H, and the absorbance peak at 1635 cm^−1^ can be assigned to the H-O-H (water molecule), indicating the presence of adsorbed water on the adsorbent [33]. The absorbance peak situated in 1445 cm^−1^ is assigned to bidentate-bicarbonate (b-HCO^3−^), b-HCO^3−^ are found to be the main surface species for the co-adsorption of CO_2_ and H_2_O on catalysis surfaces [34]. The wide peak at 1024 cm^−1^ is attributed to the S=O of SO_4_ [35]. The bands are speculated that due to the existence of bidentate bicarbonate, the material cannot degrade the adsorbed substances. After adding sulfuric acid acidification, S=O appears on the surface of the material, while bidentate bicarbonate disappears, and the adsorption sites on the surface of the adsorbent are exposed, so the material has the ability to adsorb and degrade organic substances. Under the 1000 cm^−1^ are assigned to the metal belonging to CeAY, such as the bands of 843 cm^−1^ and 575 cm^−1^ are assigned to the Ce-O, which can indicate that CeO_2_ is successfully loaded on the acid clay [36,37].

An XPS measurement was performed to investigate the Ce composition in the CeAY catalyzer. Appendix A shows the composition of Ce and O of CeO_2_, as depicted in Figure 2a, two multiplets composed of Ce 3d Spectra, which were respectively corresponding to the spin–orbit coupling of 3d_5/2_ and 3d_3/2_. Six peaks at 916.5 eV, 907.1 eV, 900.7 eV, 898.2 eV, 888.6 eV, and 888.2 eV are detected, which belong to the binding energies of Ce^4+^ 3d [38,39,40,41,42]. It should be recognized that pure ceria contained a low percentage of Ce(III) oxide, as indicated by the presence of peaks at 903.3 eV and 884.5 eV, which corresponded to the Ce 3d_3/2_ and Ce 3d_5/2_ components of Ce(III) [43,44]. One can see that CeO_2_ has the Ce^4+^ stable valence state (79.66%) and a small part of Ce^3+^ (27.34%), thus CeO_2_ has peroxidase-like activity and weak oxidase activity (Appendix A).

The survey spectrum of CeAY in Appendix A show the presence of Mg, Na, Ce, O, and Si. As depicted in Figure 2b, after the CeO_2_ was added to the acid clay, four peaks at 916.65 eV, 908 eV, 901 eV, and 898.78 eV were detected, which can be specified as the binding energies of Ce 3d_5/2_ and Ce 3d_3/2_ of Ce^4+^ [44,45,46]. Compared with Figure 2a, where the valence state of Ce^3+^ was appear at 912.5 eV, 903.9 eV, 885.5 eV, and 882.15 eV [47,48,49], the ratio of Ce^3+^/(Ce^3+^ + Ce^4+^) in the CeAY sample was increased to 51.15%. Ce mainly exists in the states of Ce^3+^ and Ce^4+^ in the catalyst. The Ce^3+^/Ce^4+^ redox pair in the catalyst would speed up the storage and release of reactive oxygen species and enhance the catalyst redox capacity due to the coexistence of Ce^3+^ and Ce^4+^ status [50]. In addition, the quantity of oxygen vacancies on the CeO_2_ catalyst is related to the ratio of Ce^3+^/(Ce^3+^ + Ce^4+^) [51]. The increase in the ratio of Ce^3+^/(Ce^3+^ + Ce^4+^) led to the formation of unsaturated chemical bonds and oxygen vacancies, keeping the active ingredient of CeAY, in the higher valence state and promoting the oxidation of the organic compound [52]. Previous studies have shown that the Ce^3+^/Ce^4+^ redox pair has a strong relationship with the oxygen holes present in the catalyst, and the oxygen holes can immediately participate in the oxidation process [53]. These flexible valence states of Ce^3+^ and Ce^4+^ prompt the CeAY to show a good charge transfer ability; the CeAY has a stronger ability to store and release reactive oxygen species; thus, the CeAY has a much stronger ability to degrade organic matter.

In order to further detect the species and vacancies of oxygen on the surface of the synthesized catalyst, the O1S was investigated (Figure 2c,d). The O1S of the XPS spectrum was decomposed into two parts, including saturated lattice oxygen (situated around 529.70 eV) and surface-absorbed oxygen species (situated around 531.30 eV) [54]. The CeAY surface-absorbed oxygen species concentration was dominantly higher than that of CeO_2_, and after the CeO_2_ was added to the acid-treated clay, the saturated lattice oxygen completely disappeared. The binding energy of 530.80 eV on CeAY is assigned to the over-layer of oxygen, that is, adsorbed oxygen [55], and the binding energy of 532.3 eV is assigned to the chemisorbed oxygen [56]. The two O1s peaks show that the surface of the CeAY is full of adsorbed oxygen, and sufficient oxygen vacancies increase the oxidase activity of the CeAY. After the CeO_2_ was added to the acid-treated clay, the binding energy of 532.35 eV was also assigned to the adsorbed oxygen. The binding energy appearance of 533 eV on acid-treated clay can be attributed to Si-O-Si, indicating that the main component of acid-treated clay is SiO_2_ (Appendix A) [57].

### 3.2. Catalytic Activity and Degradation Performance

The catalytic performance of CeO_2_, acid-treated clay, and CeAY was evaluated by the degradation of Rhoda mine B (Rh B) at room temperature (Figure 3a and Appendix A). At the condition of pH = 7.0, the Rh B was not degraded after adding the CeO_2_ and CeAY (Curve a, e, g Appendix A), and the acid-treated clay adsorbs Rh B to the bottom of the solution (Curve c, Appendix A), while at the condition of pH = 1.0, the Rh B itself will not degrade due to the decrease in pH (Curve b), the Rh B was degraded only after the addition of CeAY (Curve h, Appendix A), and there was no change in other solutions. After a week of reaction, the color of the solution with CeO_2_ was nearly colorless, after two weeks of reaction, the Rh B was completely degraded (Curve f, Appendix A), indicating that the CeO_2_ has low oxidase activity at the condition of pH = 1.0. The acid-treated clay was adsorbed in the Rh B at the beginning of the reaction, and the color of the Rh B was never changed, indicating that the acid-treated clay has only adsorption performance and no catalytic activity (Curve d) and the CeAY has high oxidation activity at the condition of pH = 1.0 (Curve h, Appendix A).

To investigate the peroxide-like and oxide-like properties of CeAY, UV-vis spectrophotometry was carried out using H_2_O_2_ and 3,3′,5,5′-tetramethylbenzidine (TMB) as the substrates. In Figure 3c, when pH was 1.0 and 4.0, respectively, the H_2_O_2_ with TMB and TMB itself did not become blue and yellow color or produce UV–vis absorbance. The CeAY can catalyze the oxidation of TMB, exhibiting a blue reaction at the condition of pH = 4 and exhibiting a yellow reaction at the condition of pH = 1, the absorption peak is 652nm and 450 nm, and after the H_2_O_2_ was fed into the solution, the absorption peak and color were strengthened. The results showed that the CeAY has peroxide-like and oxide-like activity, and the H_2_O_2_ can enhance the activity of the material.

Additionally, the influence factors such as O_2_ and temperature on the catalytic activity of CeAY were further considered. As shown in Figure 3d, the catalytic activity of the CeAY was increased with the temperature increasing and reached its highest at a temperature of 60 °C. After adding O_2_ and Ar_2_, respectively, for 40 min, UV-vis spectrophotometry was carried out using H_2_O_2_ and TMB as the substrates. As depicted in Appendix A, the absorption peak, and color were weakened when TMB was oxidized, exhibiting that O_2_ will not enhance the reaction and that Ar_2_ will not weaken the reaction. Additionally, the CeAY showed high catalytic activity and good thermal stability in a wide pH and temperature range. It is interestingly found that the CeAY will only degrade under the condition of pH = 1.0, while at the condition of pH = 2.0, the CeAY will not degrade the organic compound.

To study the catalytic mechanism of the oxidase-like activities of CeAY, the steady-state kinetic factors of the reaction were studied under optimal pH (1.0) and temperature (60 °C). The initial velocity was estimated by the slopes of the linear portion of the absorbance–time curve for TMB-derived oxidation products. The studied curves were calculated with varying concentrations of TMB as the substrates. As indicated in Figure 3e, the initial reaction rates showed obvious dependence on the substrate concentration. In addition, the Lineweaver–Burk plot was used to study the enzymatic activity of CeAY. As can be seen in Figure 3f, the reciprocal of the initial rate showed an obvious linear relationship against the reciprocal of the substrate concentration. The catalytic parameters (*K_m_* and *V_max_*) were calculated based on the double reciprocal of the Michaelis–Menten equation. The comparison between the parameters of CeAY and those of other oxidase mimics is presented in Appendix A. As shown, the as-prepared CeAY exhibited lower *K_m_* values of 0.34 mM compared to other oxidase mimics such as CeO_2_ nanoceria, indicating that maximal activity of CeAY could be achieved at a lower TMB concentration. The above results proved that the as-prepared CeAY had excellent oxidase-like activity [58,59,60,61,62].

To further assess the degradation efficacy of CeAY in actual contaminants, the degradation ability of CeAY was investigated by degrading Rhodamine B (Rh B), methyl orange (MO), aniline, and benzoic acid. The Rh B and MO have the maximum absorption peaks at 556 nm and 452 nm, respectively, which can make it easier to detect whether they are degraded. In order to further study the ability of CeAY to degrade organic pollutants, aniline, an organic pollutant in textile printing and dyeing wastewater, and benzoic acid, a degradation intermediate product of Rh B [63], were selected. As depicted in Appendix A, one can see that after the catalyst was introduced in the sewage, the Rh B (maximum absorption wavelength is 552 nm) and MO (maximum absorption wavelength is 452 nm) completely disappeared (Appendix A), and the HPLC showed that aniline was completely degraded (Figure 3b), and the benzoic acid was degraded by 50% (Appendix A).

The COD test of Rh b and MO is used to evaluate the carbon mass balance in the degradation process. From Appendix A, we can see that after 10 min of degradation, the color of MO and Rh B completely disappeared, and the COD concentration of MO almost did not change, while the COD concentration of Rh B decreased from 77 mg/L to 52mg/L, indicating that Rh B and MO were degraded into various degradation intermediate products and small molecular organic acids (Appendix A), as well as further degradation into H_2_O and CO_2_. The reason why the COD concentration of MO did not decrease may be that the N=N bond was relatively difficult to break, requiring a longer reaction time [64].

From the degradation experiment, it can be seen that 10 mg/L of aniline can be completely degraded in 10 min, meeting the effluent standard of aniline wastewater in urban sewage treatment plants. In total, 100 mg/L of benzoic acid can be degraded by 50%, proving that 50 mg/L benzoic acid wastewater can be completely degraded, meeting the wastewater discharge standard of toluene in urban sewage treatment plants. Since benzoic acid is the intermediate product of Rh B degradation, it is proven that the dye can also be completely degraded. Therefore, as long as enough materials are added, the effluent standard of each organic acid in China’s research will be met (*Discharge standard of pollutants for urban wastewater treatment plant* GB18918-2002).

In order to investigate the degradation effect of the CeAY under different conditions, a degradation experiment at the condition of pH = 7.0 and 1.0 was carried out. As depicted in Appendix A, one can see that the Rh B and MO showed no change, and after the strong acid was added into the mixed solution, the color of the Rh B and MO completely disappeared, thus proving that the material can only degrade organic matter at the condition of pH = 1.0. Furthermore, to verify the strong acid do not have an effect on Rh B and MO, the HCl and H_2_SO_4_ were added into Rh B and MO without the material, from Appendix A we can see that the absorption peak at 552 nm and 450 nm was no change, it is proved that the strong acid will not affect the dye.

### 3.3. Mechanism Study

In order to go further to explore the mechanism of the Rh B degradation via CeAY activation, free radical quenching experiments by PBQ, TBA, and L-His, which were acknowledged as effective scavengers for •OH, •O_2_^−^ and ^1^O_2_ in the degradation of Rh B, the results showed that •OH and •O_2_^−^ were the dominant active oxygen species in CeAY mediated Rh B degradation under the condition of pH = 1.0, and ^1^O_2_ showed less important roles in the degradation.

In order to further verify the oxygen vacancy, the electron paramagnetic resonance (EPR) was then performed, which provided evidence of surface defects and trapped electrons in the test sample. As shown in Figure 4b, the CeAY samples show an EPR signal at g = 2.003, which can be attributed to the electrons captured on oxygen vacancy [65]. In order to detect the existence and species of ROS, the ESR spin trap 5,5-dimethyl-1-pyrroline-n-oxide (DMPO) was used to capture free radicals, because the spin trap can capture ROS to form DMPO-ROS adducts. As shown in Figure 4c, an ESR spectra with a typical signal of 1:2:2:1 appeared, which showed that •OH was generated for CeAY. Additionally, the spectra with a signal ratio of 1:1:1:1:1:1 can be observed, and the peak intensity at positions 3 and 5 is lower, which can be attributed to the generation of •O_2_^−^ [66]. Therefore, the degradation of CeAY is attributed to its production of •OH and •O_2_^−^. Because ROS are highly reactive and nonselective, they can oxidize and decompose a variety of harmful compounds into CO_2_ and inorganic ions.

As shown in Figure 4d, we can see that after the degradation of the organic compound, two peaks at 900.8 eV and 882.5 eV can be assigned to the binding energies of Ce 3d_5/2_ and Ce 3d_3/2_ of Ce^4+^ [67,68]. Additionally, the other two peaks at 905.1 eV and 886.65 eV are assigned to the binding energies of Ce 3d_5/2_ and Ce 3d_3/2_ of Ce^3+^ [69,70]. After the degradation of the MO, the ratio of Ce^3+^/(Ce^3+^ + Ce^4+^) was increased to 74.48%. The presence of HCL and H_2_SO_4_ leads to an increase in the amount of Ce^3+^ on the catalyst surface and a decrease in the amount of Ce^4+^. It is generally believed that Ce^3+^ will cause an imbalance of electrons, thus forming the oxygen vacancies and an unsaturated chemical bond, which are conducive to the formation of surface catalysis chemisorbed oxygen [27]. The chemisorbed oxygen and oxygen vacancies will create the superoxide species for the degradation of the organic compounds [71]. As depicted in Appendix A, compared with CeAY (Figure 2d), after the degradation of the MO, the electric transfer on O1S happens, leading to the increase in binding energy, it is proved from the side that the e^-^ transformed to absorbed oxygen.

A possible catalytic mechanism was proposed in Figure 5, where the existence of Na_3_C_6_H_5_O_7_•2H_2_O enhanced the reduction of CeO_2_ to form a 20–30 nm sized microspheres. The CeO_2_ microsphere, having an average diameter of 6.22 nm, was well dispersed due to the presence of activated clay in the medium. In an XPS study, we showed that the reduction of Ce^4+^ proceeded through the hydrothermal process with Na_3_C_6_H_5_O_7_•2H_2_O, and during the degradation, the reduction of Ce^4+^ was to some extent due to the presence of H^+^. Further, after the addition of strong acid, the Ce^3+^ allows for the fast transfer of electrons generated in the CeAY framework to adsorbed O_2_, producing •O_2_^−^. Meanwhile, the high concentration of H^+^ in the reaction system also promoted the conversion of •O_2_^−^ to H_2_O_2_, the Ce^3+^ catalyzes H_2_O_2_ to •OH, and then •O_2_^−^ and •OH readily contribute to the degradation of organic compounds to form CO_2_ and H_2_O as an oxidation product. CeAY prepared by CeO_2_ and activated clay can be a good catalyst for removing organics to overcome the wastewater treatment problem of strong acid industries.

## 4. Conclusions

This work developed a method for highly efficient O_2_ activation and utilization for organic degradation using Ce-O nanosphere as catalysts and surface-associated Ce^3+^ species as oxidants. Unlike the Fenton reaction systems, oxidation of organic pollutants by Ce^3+^ can break the limit of pH and H_2_O_2_. The CeAY has excellent oxidative activity, which can catalyze O_2_ to produce •OH and •O_2_^−^ to degrade organic compounds and soluble anionic-cationic dyes. It is noteworthy that the material can degrade organic compounds under the condition of pH = 1.0 or more acid, eliminate the pollution of organic matter in acidic wastewater, and recover valuable resources. It solves the problem of acid recovery caused by organic pollutants in strongly acidic wastewater, which puts forward an effective method and new idea for the degradation of strong acid industrial organic wastewater. However, according to the price survey on Macklin’s official website, the cost price of 1 g of CeAY is CNY 0.546. Additionally, the CeAY can only play a role in strongly acidic conditions. It cannot be used as a sewage treatment agent in natural water bodies or sewage treatment plants. It can only be used as a pretreatment agent for strong acidic wastewater in plants to degrade organic pollutants, reduce the treatment cost of sewage treatment plants, and facilitate acid recovery. High prices and harsh reaction conditions restrict the utilization of CeAY.

## Figures and Tables

**Figure 1 nanomaterials-12-04410-f001:**
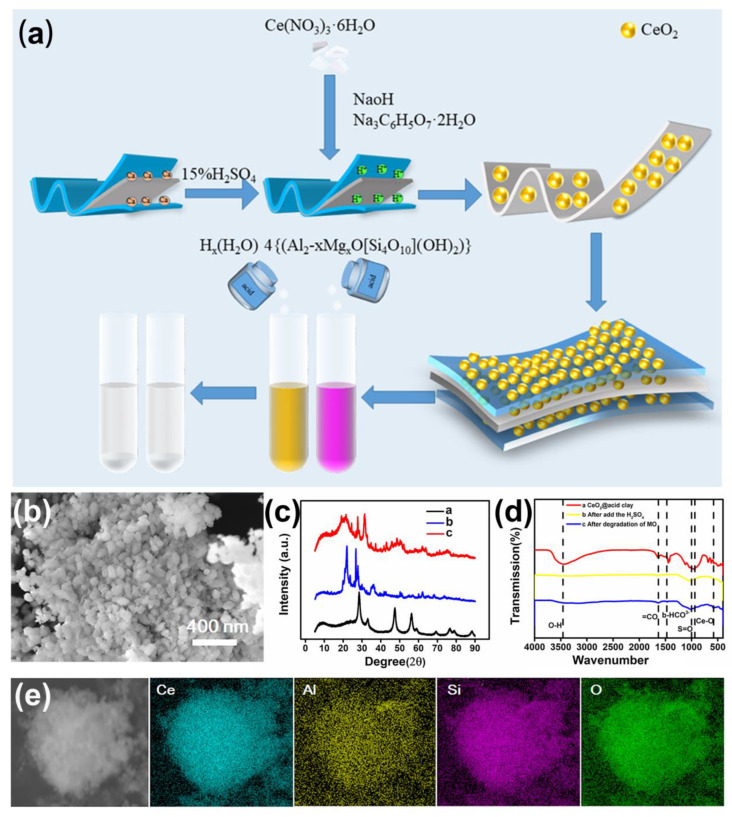
Schematic illustration of the CeAY preparation (**a**); SEM images of CeAY (**b**); XRD patterns of the comparison of three materials (**c**); the FT-IR investigates of CeAY (**a**), acid-treated clay (**b**) and CeO_2_ (**c**,**d**); EDS compositional mapping of CeAY (**e**).

**Figure 2 nanomaterials-12-04410-f002:**
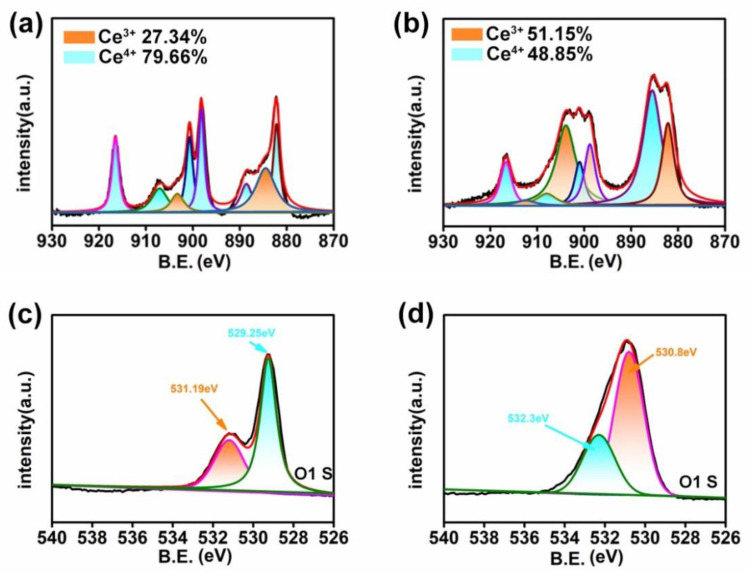
The Ce 3d XPS patterns of CeO_2_ (**a**), CeAY (**b**) and the O1s patterns of CeO_2_ (**c**), CeAY (**d**).

**Figure 3 nanomaterials-12-04410-f003:**
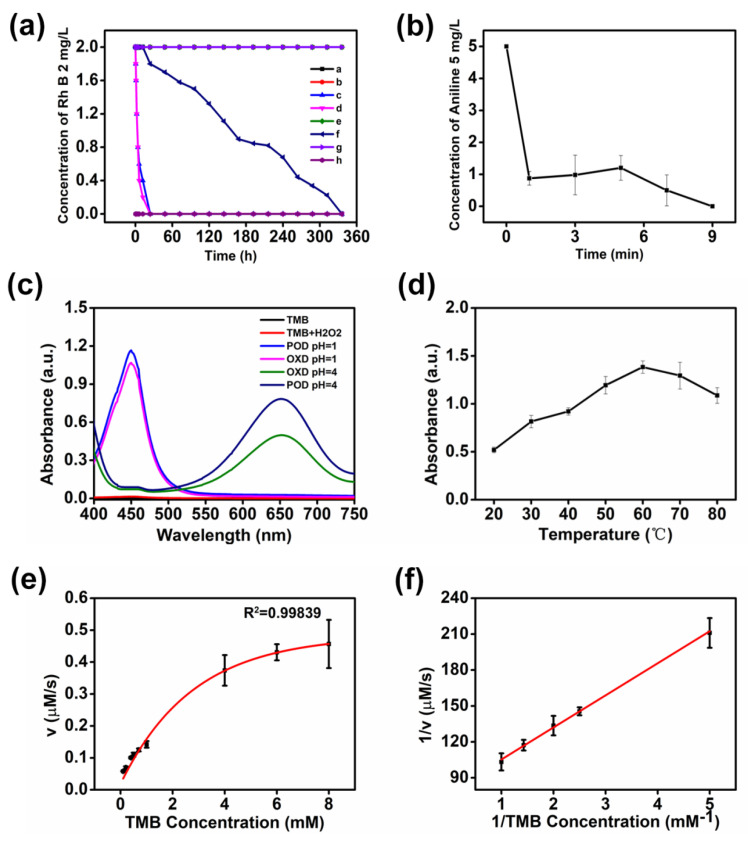
The degradation of Rh B with different materials (**a**), (a) Rh B; (b) Rh B, pH = 1.0; (c) acid-treated clay; (d) acid-treated clay, pH = 1.0; (e) CeO_2_; (f) CeO_2_, pH = 1.0; (g) CeAY; (h) CeAY, pH = 1.0; The degradation efficiency of aniline (CeAY 2 mg mL^−1^, room temperature, pH = 1, 5 mg L^−1^) (**b**); the peroxidase-like and oxidase-like activity of CeAY (0.04 mg mL^−1^, 2 mM TMB, pH = 1) (**c**); The temperature effort (0.04 mg mL^−1^, 2 mM TMB, pH = 1) (**d**); Steady-state kinetic and catalytic mechanism of the catalytic activity of the CeAY. The velocity (v) of the reaction was evaluated using 50 µL of CeAY (0.04 mg mL^−1^) in 0.1 mol L^−1^ HCl (pH = 1.0) at 60 °C. (**e**,**f**). The error bars stand for the standard deviation (*n* = 3).

**Figure 4 nanomaterials-12-04410-f004:**
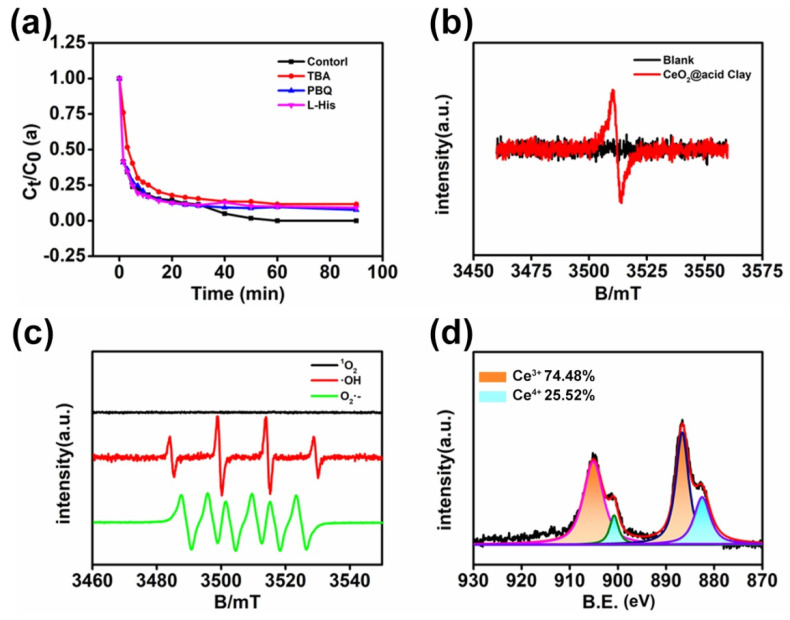
Quenching experiments over CeAY under pH = 1.0 (**a**). EPR spectra (**b**), detection of •OH and •O_2_^−^ analyzed by ESR measurement (**c**), the Ce 3d XPS patterns of CeAY after degradation of MO (**d**).

**Figure 5 nanomaterials-12-04410-f005:**
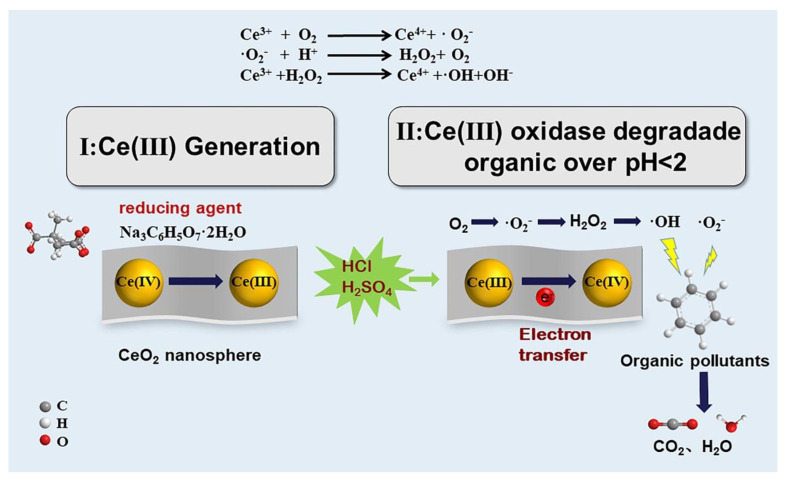
Schematic diagram of mechanism CeAY oxidizes organic in low pH and decomposition into H_2_O and CO_2_.

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
