# Peer review of "Oxygen Vacancy-Mediated Activates Oxygen to Produce Reactive Oxygen Species (ROS) on Ce-Modified Activated Clay for Degradation of Organic Compounds without Hydrogen Peroxide in Strong Acid"

_nanomaterials, 2022, doi:10.3390/nano12244410_

Round 1
Reviewer 2 Report
This paper used oxidation of CeAY (CeO2 acid clay) to remove organic pollutants in acid wastewater. Results of the study may have important application in the field of industrial wastewater treatment. Authors may wish to consider the following in revision of their manuscript.
1. Rh B、MO、Aniline and Benzoic acid were chosen in the study. Please comment on why these 4 types of organic acids were chosen.
2. Please provide cost of preparation of proposed catalyst used in the study.
3. Please provide mass balance of carbon in the degradation study.
4. Please comment on the limitations of using proposed catalysts in acid wastewater treatment.
5. Please comment on whether the proposed study can meet effluent standards for each type of organic acid used in the study in China.
6. Please compare the treatment performance of the proposed study with other studies reported in the literature using different catalysts.
Author Response
请参阅附件
